# Light reintroduction after dark exposure reactivates plasticity in adults via perisynaptic activation of MMP-9

Sachiko Murase†*, Crystal L Lantz†, Elizabeth M Quinlan*

Neuroscience and Cognitive Sciences Program, Department of Biology, University of Maryland, Maryland, United States

**Abstract** The sensitivity of ocular dominance to regulation by monocular deprivation is the canonical model of plasticity confined to a critical period. However, we have previously shown that visual deprivation through dark exposure (DE) reactivates critical period plasticity in adults. Previous work assumed that the elimination of visual input was sufficient to enhance plasticity in the adult mouse visual cortex. In contrast, here we show that light reintroduction (LRx) after DE is responsible for the reactivation of plasticity. LRx triggers degradation of the ECM, which is blocked by pharmacological inhibition or genetic ablation of matrix metalloproteinase-9 (MMP-9). LRx induces an increase in MMP-9 activity that is perisynaptic and enriched at thalamo-cortical synapses. The reactivation of plasticity by LRx is absent in $Mmp9^{-/-}$ mice, and is rescued by hyaluronidase, an enzyme that degrades core ECM components. Thus, the LRx-induced increase in MMP-9 removes constraints on structural and functional plasticity in the mature cortex.
DOI: https://doi.org/10.7554/eLife.27345.001

*For correspondence:
smurase@umd.edu (SM);
equinlan@umd.edu (EMQ)

†These authors contributed equally to this work

Competing interests: The authors declare that no competing interests exist.

## Introduction

The structural and functional plasticity revealed by monocular deprivation (MD) is highest during a postnatal critical period (*Wiesel and Hubel, 1963*). Although the loss of this plasticity with development was previously thought to be irreversible, several experimental manipulations have been identified that can reactivate critical period plasticity in adults (for review see *Hübener and Bonhoeffer, 2014*). Using rodent models, we have demonstrated that visual deprivation through dark exposure reactivates robust ocular dominance plasticity in the adult visual cortex (*He et al., 2006*). Importantly, the plasticity that is reactivated by dark exposure (DE) can be harnessed to promote the recovery from severe amblyopia (*He et al., 2007*; *Montey and Quinlan, 2011*; *Eaton et al., 2016*). The reactivation of plasticity through dark exposure has now been demonstrated in several species, including kittens (*Duffy and Mitchell, 2013*; *Mitchell et al., 2016*) and is effective in mice up to P535 days old (*Stodieck et al., 2014*). However, very little is known regarding the mechanism by which dark exposure enhances plasticity in the adult visual cortex.

### Maturation of the ECM constrains plasticity

The maturation of fast-spiking interneurons (FS INs), which mediate the perisomatic inhibition of pyramidal neurons, regulates the timing of the critical period for ocular dominance plasticity (*Hensch et al., 1998*; *Morishita et al., 2015*; *Stephany et al., 2016*; *Kuhlman et al., 2013*; *Gu et al., 2013*; *Gu et al., 2016*; *Sun et al., 2016*). The many specializations of FS INs include 1) a narrow action potential waveform, 2) a high likelihood of expressing the $Ca^{2+}$ binding protein parvalbumin ($PV^+$) and 3) a particularly dense perineuronal net (PNN), a specialization of extracellular matrix (ECM). The ECM in the central nervous system consists of a matrix of chondroitin sulfate proteoglycans (CSPGs) and hyaluronic acid, organized into PNNs by tenascins and cartilage link protein

(*Carulli et al., 2010*; *Morawski et al., 2014*; *Celio and Chiquet-Ehrismann, 1993*). Functions attributed to the mature ECM include the imposition of a physical barrier to structural plasticity, the regulation of neuronal excitability via the accumulation of cations and other effectors (*Härtig et al., 1999*; *Sugiyama et al., 2008*), the protection of neurons from oxidative stress (*Cabungcal et al., 2013*) and the sequestration of molecules that inhibit neurite outgrowth (*Stephany et al., 2016*; *Vo et al., 2013*).

The maturation of the ECM in general, and PNNs in particular, has been negatively correlated with the expression of synaptic plasticity. An enhancement of plasticity following degradation of the mature ECM was first demonstrated in rodent primary visual cortex (*Pizzorusso et al., 2002*; *Pizzorusso et al., 2006*). Degradation of the ECM and a subsequent enhancement of synaptic plasticity has now been demonstrated in many brain regions, including the amygdala (*Gogolla et al., 2009*), perirhinal cortex (*Romberg et al., 2013*) and hippocampus (*Kochlamazashvili et al., 2010*; *Carstens et al., 2016*), suggesting that the maturation of the ECM is a general mechanism for restricting change in synaptic structure and function. Furthermore, dark-rearing from birth delays the maturation of PNNs and prolongs the critical period for ocular dominance plasticity (*Pizzorusso et al., 2002*; *Mower, 1991*; *Lander et al., 1997*). Interestingly, environmental enrichment reverses these effects of dark rearing, perhaps by accelerating epigenetic maturation (*Bartoletti et al., 2004*; *Baroncelli et al., 2016*). Maturation of PNNs by environmental enrichment is also observed in the CA2 region of hippocampus (*Carstens et al., 2016*). However, in the absence of dark-rearing, environmental enrichment reduces PNN density and enhances plasticity in the primary visual cortex of normal and amblyopic adult mice (*Sale et al., 2007*; *Scali et al., 2012*; *Greifzu et al., 2014*; *Greifzu et al., 2016*).

## Constraints imposed by the maturation of the ECM are reversed by DE/LRx

Several manipulations that impact the expression of ocular dominance plasticity interfere with the maturation of PV$^+$ INs and PNNs. These include redox dysregulation of PV$^+$ INs (*Morishita et al., 2015*), genetic deletion of neuronal activity-related pentraxin (NARP; *Gu et al., 2013*) or cartilage link protein (*Carulli et al., 2010*) and inhibition of NRG1/erbB4 signaling (*Gu et al., 2016*; *Sun et al., 2016*). Furthermore, deletion of nogo-66 receptor 1 (*ngr1*), which interacts with the glycosaminoglycan (GAG) side chains of CSPGs, prevents termination of the critical period (*Dickendesher et al., 2012*; *Frantz et al., 2016*). Manipulations that reduce PV$^+$ IN spiking may enable a shift in ocular preference of pyramidal neurons by regulating neuronal spiking output, thereby mimicking the pathway engaged by monocular deprivation in juveniles (*Kuhlman et al., 2013*). A decrease in the density of PNNs specific to PV$^+$ INs is often reported as the consequence of extracellular proteolysis. However, plasticity at excitatory synapses onto PNN-bearing pyramidal neurons in hippocampal CA2, which are typically aplastic, is directly enhanced by chondroitinase (*Zhao et al., 2007*; *Carstens et al., 2016*). Furthermore, diffuse ECM surrounds all neurons, and degradation of components of the ECM would be expected to impact plasticity at synapses on both PNN and non-PNN bearing neurons.

Interestingly, many of the constraints on synaptic plasticity imposed by maturation of the ECM appear to be reversed by DE, suggesting that extracellular signaling pathways may mediate the reactivation of synaptic plasticity by visual deprivation in adulthood. Here we test this hypothesis directly, by examining the integrity of the ECM following visual deprivation in adult mice. Previous work utilizing DE predicted that the elimination of visual input was sufficient to reactivate plasticity in adult visual cortex, however, here we show that DE alone does not impact the integrity of the ECM. In contrast, light reintroduction (LRx) following dark exposure significantly degrades the ECM, increases the perisynaptic activity of matrix metalloproteinase-9 (MMP-9) and reactivates structural and functional plasticity the adult visual cortex.

## Results

### Light reintroduction (LRx) after dark exposure induces MMP-9 dependent degradation of extracellular matrix (ECM)

The distribution of chondroitin sulfate proteoglycans (CSPGs), a primary component of the ECM, can be revealed by the plant lectin Wisteria-floribunda agglutinin (WFA) which binds to a specific monosaccharide in the chondroitin sulfate chain (N-acetyl-D-galactosamine (GalNAc; (Kurokawa et al., 1976)). FITC-WFA revealed the differential distribution of CSPGs in the binocular region of mouse primary visual cortex (V1b), including diffuse labeling in white matter, and concentrated labeling around many PV$^+$ INs (Figure 1A–F). Surprisingly, we observed no change in WFA labeling intensity or distribution after 10 days of dark exposure (DE, 99.1±5.2% of control, n=6 subjects). In contrast, light reintroduction (LRx; 24 hr) following DE induced a significant decrease in the intensity of WFA labeling (74.7±3.3% of control, n=7 subjects, One-way ANOVA, $F_{(df,18)}$=13.6, p=0.00036, Tukey-Kramer post hoc: LRx versus Con p<0.01; DE versus Con p=0.2). The decrease in WFA staining was most pronounced 250 – 400 μm from the cortical surface, a region highly enriched in thalamic afferents (48.8±5.3% of control, n=5 subjects, One-way ANOVA, $F_{(df,14)}$=6.01, p=0.016, p<0.05 Tukey-Kramer post hoc, Figure 1G and H). Concurrent tracking of PV expression revealed a decrease by DE (78.4±2.3% of control, n=6 subjects) as expected for a calcium binding protein regulated by activity (Donato et al., 2013). 24 hr of LRx increased the expression of PV to control levels (88.8±6.3% of control, n=7 subjects, One-way ANOVA, $F_{(df,18)}$=11.4, *p<0.05, Tukey-Kramer post hoc LRx versus Con p<0.05; DE versus Con p=0.045). Importantly, the decrease in WFA staining following LRx was observed in both PV$^+$ and PV$^-$ regions (Figure 1I), suggesting widespread degradation of the ECM. Similar results were obtained by tracking ECM integrity with an antibody against aggrecan, an abundant CSPG in the adult ECM (Lau et al., 2013). Aggrecan immunoreactivity was unchanged by dark exposure and significantly decreased by LRx (Figure 2A).

Aggrecan, and other components of the extracellular environment, are substrates for the activity-dependent matrix metalloproteinase MMP-9 (Mercuri et al., 2000). Therefore, to ask if degradation of ECM was dependent on the activity of MMP-9, we examined the response to LRx in the presence of a potent MMP-9 inhibitor (Inhibitor I, CAS 1177749-58-4; IC$_{50}$=5 nM). In vivo infusion of the MMP-9 inhibitor (5 nM, 100 μl via micro-osmotic mini pump, beginning 6 days prior to LRx, with concurrent DE) prevented the degradation of ECM by 24 hr of LRx (Figure 2A and B). Quantitative immunoblotting of visual cortex homogenate revealed a significant decrease in active MMP-9 following DE and a significant increase following LRx (Figure 2—figure supplement 1A). The rapid activation of MMP-9 appears to be a result of visual experience, as the level of active MMP-9 in medial prefrontal cortex was unchanged in the same subjects (Figure 2—figure supplement 1B).

### LRx induces an increase in MMP-9 activity at thalamo-cortical synapses

MMP-9 is relatively promiscuous, therefore the consequence of MMP-9 activity is highly dependent on the location of enzyme activation. To identify the location of enhanced MMP-9 activity by LRx, we employed in vivo zymography, the use of an exogenous enzyme substrate (dye-quenched gelatin; D12054, ThermoFisher Scientific, 2 mg/ml, excitation/emission=495/519 nm) in which fluorescence emission is blocked by intramolecular quenching. Proteolysis of the substrate relieves the quenching, such that fluorescence emission reports enzymatic activity. Because MMP-9 has overlapping substrate specificity with MMP-2 (Szklarczyk et al., 2002), the zymography substrate (hereafter called MMP biomarker) cannot distinguish between the activities of these two related metalloproteinases.

Delivery of the MMP biomarker (2 mg/ml, 4 μl at 100 nl/min, via cannula implanted 3 weeks prior to injection) was well tolerated, with no evidence of enhanced reactivity of astrocytes or microglia (staining for glial fibrillary acidic protein (GFAP) and Iba1, Figure 3—figure supplement 1). In vivo delivery of the biomarker 24 hr before analysis reveals that MMP activity is punctate in the adult V1 (~1.1 μm$^2$ diameter), but puncta density and intensity are relatively low (Figure 3). Ten days of DE did not significantly change biomarker puncta size, density, or intensity. In contrast, LRx induced a two-fold increase in the density and intensity of fluorescence, although puncta size was unchanged (Figure 3A). Biomarker puncta co-localized with markers for excitatory axons from cortical neurons (vesicular glutamate transporter 1, VGluT1) and thalamic neurons (vesicular glutamate transporter 2, VGluT2) in adult V1. Importantly, LRx induced an increase only in biomarker co-localization with

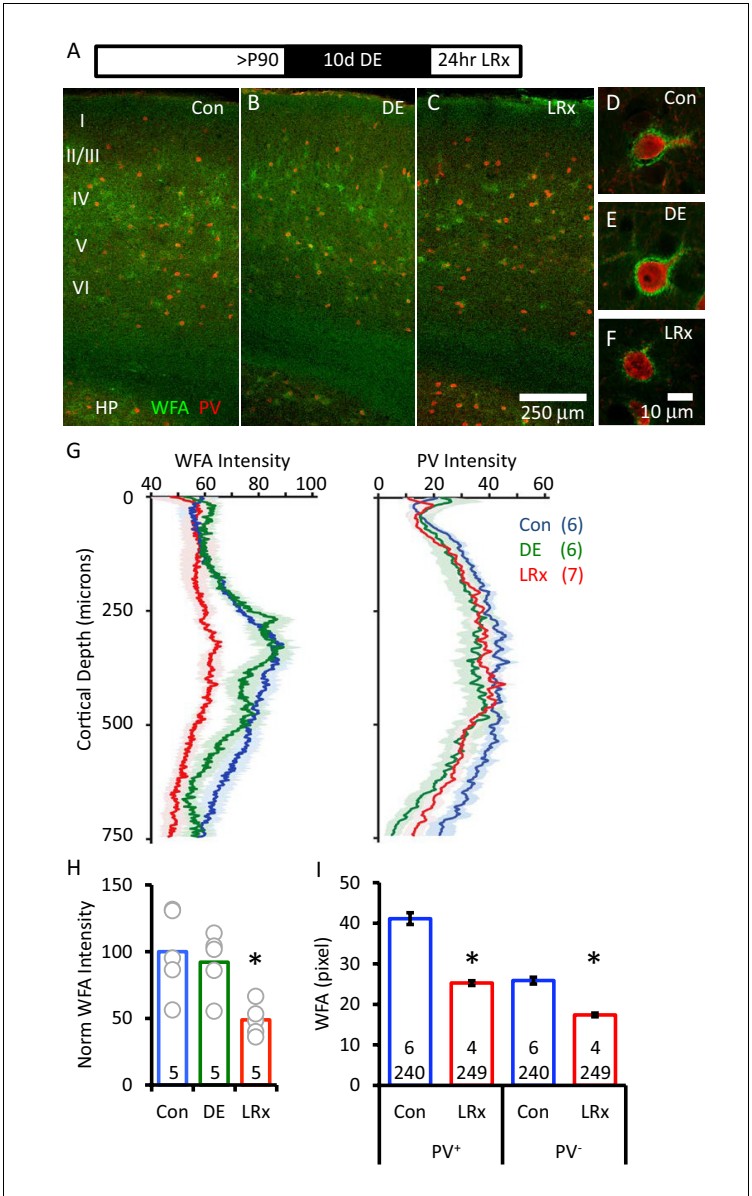

**Figure 1.** Degradation of ECM in primary visual cortex by light reintroduction (LRx). Top: Experimental paradigm. C57B/L6J mice raised in 12/12 hr light/dark cycle until postnatal day 90 (P90) received 10 days of dark exposure (DE) with subsequent light reintroduction (1 day; LRx). (**A-C**) Low magnification (10X) double labeled images of the binocular region of primary visual cortex with FITC-wisteria floribunda agglutinin (WFA; green) and Alexa-546 anti-parvalbumin antibody (PV; red) in control subjects (**A**), following dark exposure (**B**) and following dark exposure/light reintroduction (**C**). Approximate locations of layers I to VI and hippocampus (HP) are shown. (**D-F**) High magnification images (100X). (**G**) Quantification of mean fluorescence intensities for WFA (left) and PV (right) in maximal intensity projections (Z-stack 3 × 7.5 µm x of an area 450 µm wide x 750 µm deep; spanning all cortical layers). Mean±SEM; n=6, 6, 7 subjects for Con, DE, LRx, respectively. One way ANOVA, WFA; F=13.57, p=0.0004. PV; F=3.79, p=0.045. (**H**) Quantification of WFA intensity in region of interest 250 – 400 µm from surface, normalized to average control (n=5 subjects each). One-way ANOVA, F=6.01; p=0.0016. *p<0.05, Tukey-Kramer *post hoc*. (**I**) LRx induced a decrease in WFA intensity in PV+ and PV- pixels. (Control 6 subjects, 240 ROIs) versus LRx (4 subjects, 249 ROIs); *p<0.005, Student's T-test.

DOI: https://doi.org/10.7554/eLife.27345.002

The following source data is available for figure 1:

**Source data 1.**

DOI: https://doi.org/10.7554/eLife.27345.003

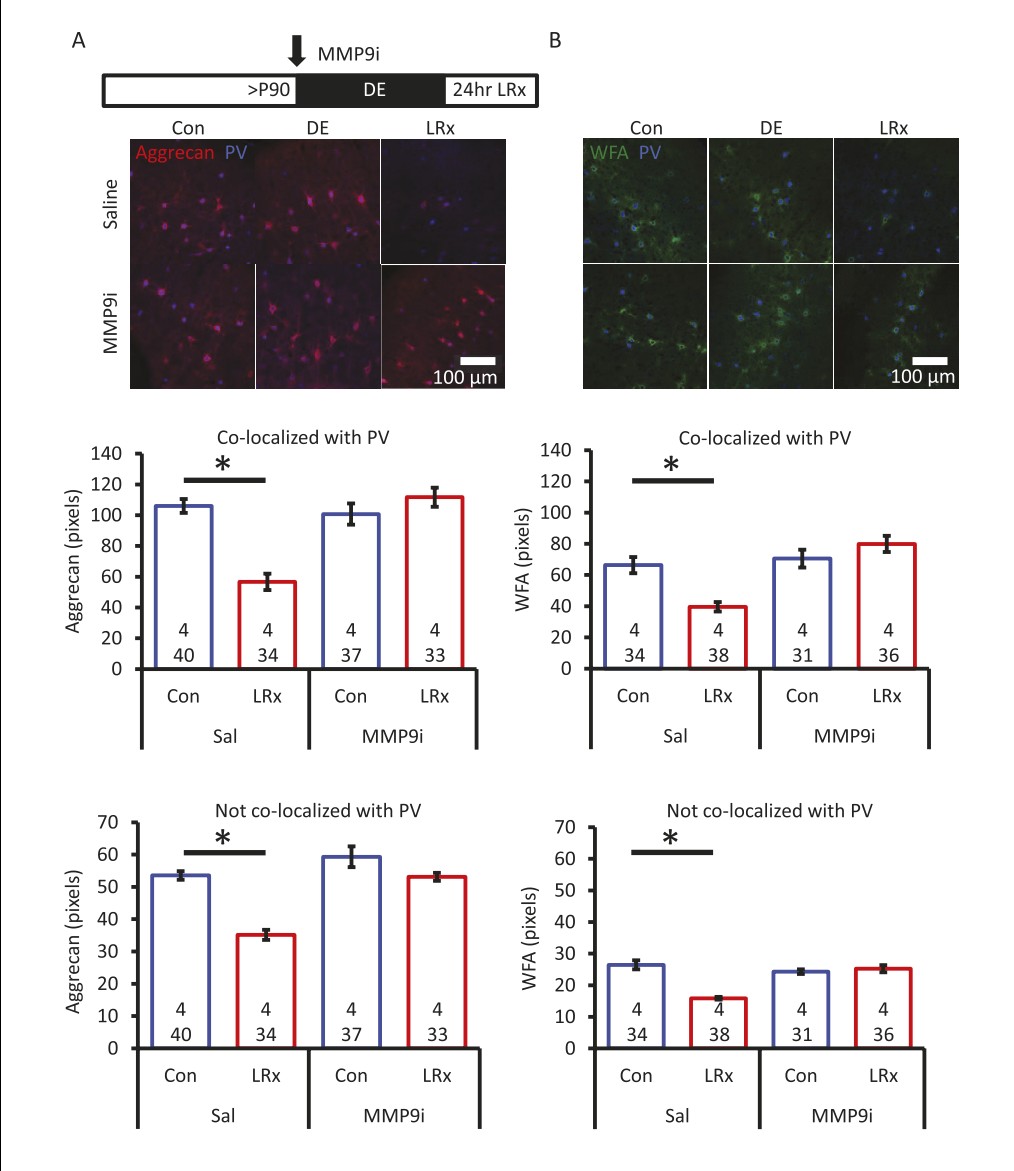

**Figure 2.** Degradation of ECM by light reintroduction is blocked by MMP-9 inhibitor. Inset: experimental paradigm. MMP-9 inhibitor (MMP9i; 5 nM) was delivered *i.c.* via Alzet 1007D micro-osmotic mini pumps 6 days prior to LRx, concurrent with DE. (**A**) Top: Double label confocal micrographs of anti-aggrecan (red) and anti-parvalbumin immunoreactivity (blue) in each experimental condition. Bottom: Population data. Aggrecan intensity measured in the region 250 – 400 μm from cortical surface. One-way ANOVA, Aggrecan, co-localized with PV: F=18.81, p<0.0001, not co-localized with PV: F=26.32, p<0.0001. *p<0.01, Tukey-Kramer *post hoc*. (**B**) Top: Double label confocal micrographs of WFA staining (green) and anti-parvalbumin immunoreactivity (blue) in each experimental condition. Bottom: Population data. WFA intensity measured 250 – 400 μm from cortical surface. One-way ANOVA, WFA, co-localized with PV: F=12.92, p<0.0001, not co-localized with PV: F=22.49, p<0.0001. *p<0.01, Tukey-Kramer *post hoc*. n=(subjects, cells).

DOI: https://doi.org/10.7554/eLife.27345.004

The following source data and figure supplements are available for figure 2:

**Source data 1.**
DOI: https://doi.org/10.7554/eLife.27345.007
**Figure supplement 1.** Rapid increase in active MMP-9 in V1 by light reintroduction.
DOI: https://doi.org/10.7554/eLife.27345.005
**Figure supplement 1—source data 1.**
DOI: https://doi.org/10.7554/eLife.27345.006

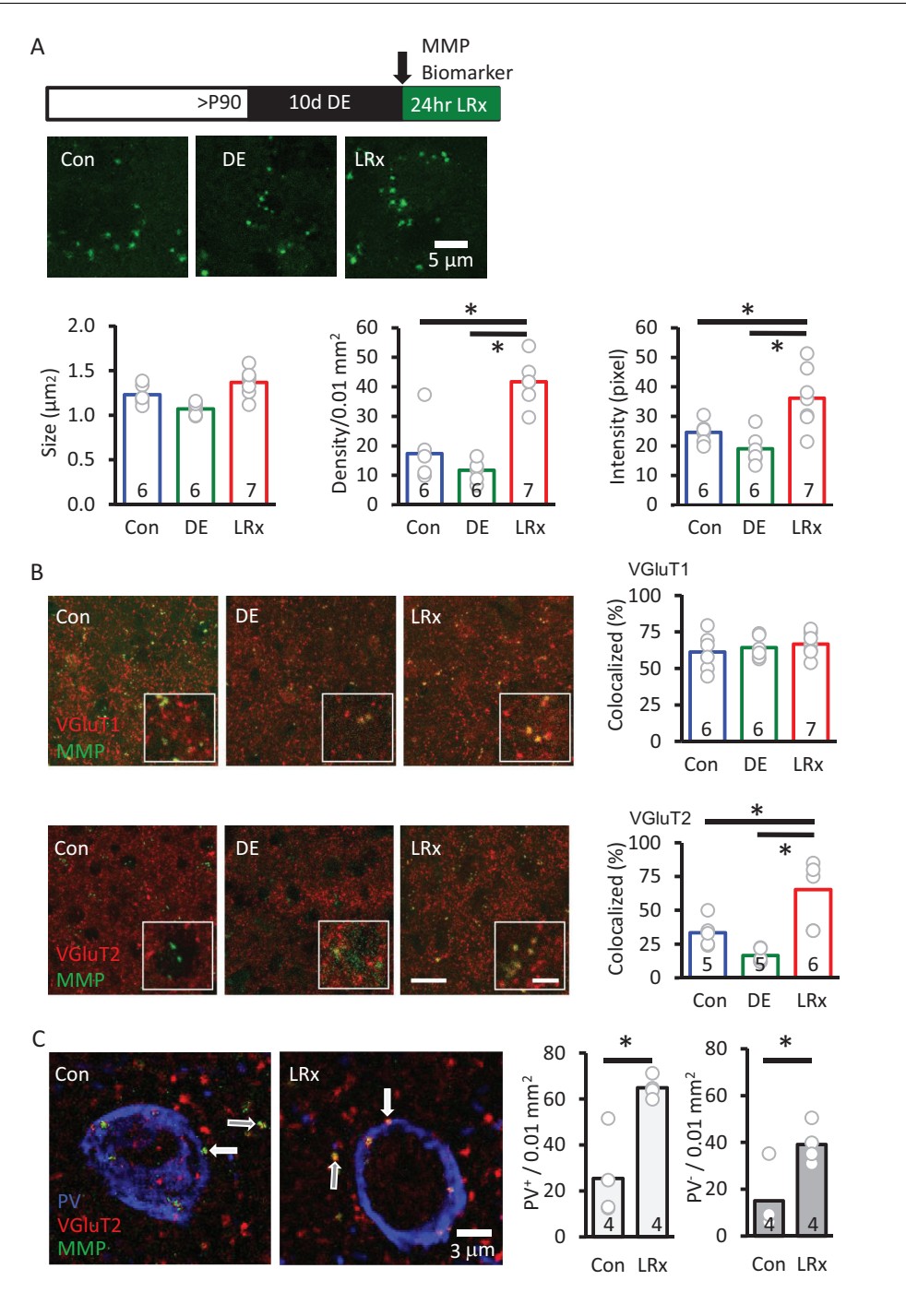

**Figure 3.** MMP biomarker reports in vivo proteolyis at synapses following LRx. Inset: experimental paradigm. The MMP biomarker, dye-quenched gelatin (2 mg/ml), was delivered *i.c.* via cannula at the initiation of LRx. (**A**) The biomarker reveals punctate staining, and a significant increase in density and intensity following LRx (n=6, 6, 8 subjects for Con, DE, LRx, respectively). One-way ANOVA, F=9.2; p=0.002 for intensity, F=27.74; p<0.0001 for density, F=10.17, p=0.0014 for size; *p<0.05, Tukey-Kramer *post hoc*. (**B**) Double labeled confocal micrographs of control, DE and LRx visual cortex labeled with MMP biomarker (green) and marker for cortical axons (top: VGluT1; red) or thalamic axons (bottom: VGluT2; red). Scale bars: 20 µm (insets: 5 µm). Co-localization with VGluT2, but not VGluT1, is significantly increased by LRx. One-way ANOVA, F=0.95, p=0.41 for VGluT1 (n=6, 6, 7 subjects for Con, DE, LRx, respectively). F=16.16, p=0.0003 for VGluT2 (n=5, 5, 6 subjects for Con, DE, LRx, respectively). *p<0.05, Tukey-Kramer post hoc. (**C**) Increase in co-localization of MMP biomarker/VGluT2 with PV+ and PV-

*Figure 3 continued on next page*

*Figure 3 continued*

immunoreactive puncta following LRx (n=4 subjects each for Con and LRx). *p<0.005, Student's T-test. All quantifications were performed 250 – 400 μm from the cortical surface in V1b.

DOI: https://doi.org/10.7554/eLife.27345.008

The following source data and figure supplement are available for figure 3:

**Source data 1.**

DOI: https://doi.org/10.7554/eLife.27345.010

**Figure supplement 1.** Expression of markers of reactive astrocytes (GFAP) and microglia (Iba1) inV1b following single cortical delivery of saline, MMP-9 biomarker, hyaluronidase and active MMP-9.

DOI: https://doi.org/10.7554/eLife.27345.009

VGluT2 (*Figure 3B*), and line scans of fluorescent triple-labeled images revealed that this reduction was observed for both PV⁺ and PV⁻ puncta (*Figure 3C*).

## Absence of response to light reintroduction in visual cortex of *Mmp9*⁻/⁻ mouse

To further explore the regulation of ECM integrity by visual experience, we examined the effect of LRx in *Mmp9* null mutant mice (*Mmp9*⁻/⁻). WFA staining revealed that deletion of *Mmp9* did not alter the distribution of CSPGs, including the diffuse distribution through all cortical layers, concentration into PNNs around many PV⁺ INs, and enrichment in the thalamo-recipient zone (*Figure 4A–F*). However, in the absence of *Mmp9*, we observed no change in ECM following DE or LRx (*Figure 4G*, one-way ANOVA, $F_{(df,17)}=0.21$, p=0.81, p=0.2 Tukey-Kramer *post hoc*, n=5, 6, 7 Con, DE, LRx). As expected, biomarker puncta persisted in the visual cortex of *Mmp9*⁻/⁻ mice, reflecting the activity of MMP-2. These biomarker puncta were reduced in size, density and intensity relative to WT (*Figure 4H*), consistent with low basal levels of MMP-2 activity in mouse cortex (*Wilczynski et al., 2008*). However, we observed no effect of LRx on biomarker puncta size, density or intensity in the absence of *Mmp9* (*Figure 4H*). Together, this indicates that the degradation of the ECM following LRx is mediated by MMP-9.

## Absence of LRx-induced changes in excitability and synchrony in *Mmp9*⁻/⁻ mice

Reduction in the inhibitory output of FS basket cells has been shown to reactivate plasticity in the adult visual cortex (*Kuhlman et al., 2013*; *Kaplan et al., 2016*). To ask if LRx regulates neuronal excitability (i.e. spiking output), we examined neuronal responses in vivo using chronically-implanted microelectrode arrays in awake, head-fixed WT and *Mmp9*⁻/⁻ mice. Single unit waveforms sorted into two populations representing fast-spiking interneurons (FS INs; presumptive PV⁺ INs) and regular spiking neurons (RS; presumptive pyramidal neurons; *Figure 5—figure supplement 1A,B*) similarly in *Mmp9*⁻/⁻ and WT mice. In adult WTs, LRx (6 hr) induced a significant decrease in the visually-evoked response in FS INs (100% contrast, 0.05 cycle per degree gratings, reversing at 1 Hz, stimulus onset=time 0 s; n=8 subjects, 16 units, p=0.02083, unpaired Student's T-test) and a significant increase in the response of RS neurons (n=8 subjects, 45 units, p=0.000209, unpaired Student's T-test; *Figure 5A*). A similar decrease in FS IN excitability and increase in RS excitability was observed in spontaneous spike rates following LRx (*Figure 5—figure supplement 1C*). As the synchronous activity of FS INs is implicated in the generation of neuronal oscillations, especially in the gamma frequency (30–120 Hz), we asked if LRx changed the distribution of power across oscillatory frequencies ranging from 0 to 120 Hz. LRx induced a decrease in the power of high γ (60–120 Hz, n=9 subjects, p=0.0168, paired Student's T-test) and an increase in the power of δ (1–4 Hz, n=9 subjects, p=0.01903, paired Student's T-test). However, we observed no change in the excitability of FS INs (n=5 subjects, 6 units) or RS neurons following LRx (n=5 subjects, 25 units; *Figure 5A*). As we have previously reported, visually evoked, but not spontaneous activity was lower in FS INs and higher in RS neurons in *Mmp9*⁻/⁻ mice (*Figure 5—figure supplement 1B* (*Murase et al., 2016*). In addition, LRx did not regulate oscillatory power in *Mmp9*⁻/⁻ mice (n=6 subjects; *Figure 5B*). However, high contrast, low spatial frequency visual stimuli revealed difference in VEP amplitudes, consistent with the assertion that feed-forward thalamic input to the cortex is an important site of MMP-9

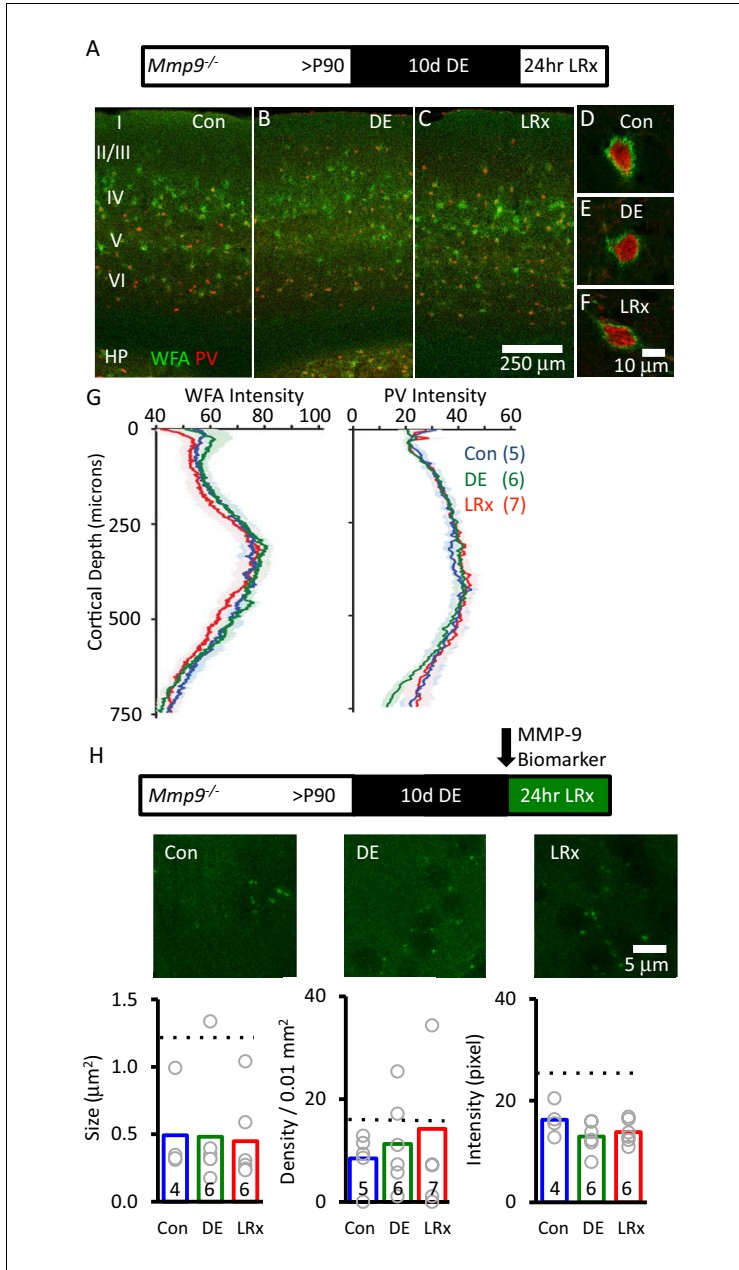

**Figure 4.** Absence of ECM degradation by light reintroduction in $Mmp9^{-/-}$ mice. Top Inset: Experimental paradigm. $Mmp9^{-/-}$ mice (P90) raised in 12/12 hr light/dark cycle until postnatal day 90 (P90) received 10 days of dark exposure (DE) with subsequent reintroduction to light (1 day; LRx). (A-C) Low magnification (10X) double labeled images of binocular region of primary visual cortex with FITC-wisteria floribunda agglutinin (WFA; green) and Alexa-546 anti-parvalbumin antibody (PV; red) in normal reared control (A) following dark exposure (B) and following dark exposure/light reintroduction (C). Approximate locations of layer I to VI, and hippocampus (HP) are shown. (D-F) High magnification images (100X). (G) Quantification of mean fluorescence intensities for WFA (left) and PV (right) in maximal intensity projections (Z-stack 3 × 7.5 µm in an area 450 µm wide x 750 µm deep; spanning all cortical layers). Mean±SEM (n=5, 6, 7 subjects for Con, DE, LRx, respectively). One way ANOVA for WFA, F=0.21, p=0.81; for PV F=0.25, p=0.78. (H) Top: experimental paradigm. The MMP biomarker, dye-quenched gelatin (2 mg/ml), was delivered i.c. via cannula at the initiation of LRx. No change in MMP biomarker puncta size (n=4, 6, 6 subjects for Con, DE, LRx, respectively), density (n=5, 6, 7 subjects) or intensity (n=4, 6, 6 subjects) was observed following DE or LRx in $Mmp9^{-/-}$ mice. Dotted lines show values for the wild type controls presented in **Figure 3**. One-way ANOVA, F=0.02, p=0.98 for size F=1.92; p=0.724 for density; F=0.33; p=0.186 for intensity.

*Figure 4 continued on next page*

*Figure 4 continued*
DOI: https://doi.org/10.7554/eLife.27345.011
The following source data is available for figure 4:
**Source data 1.**
DOI: https://doi.org/10.7554/eLife.27345.012

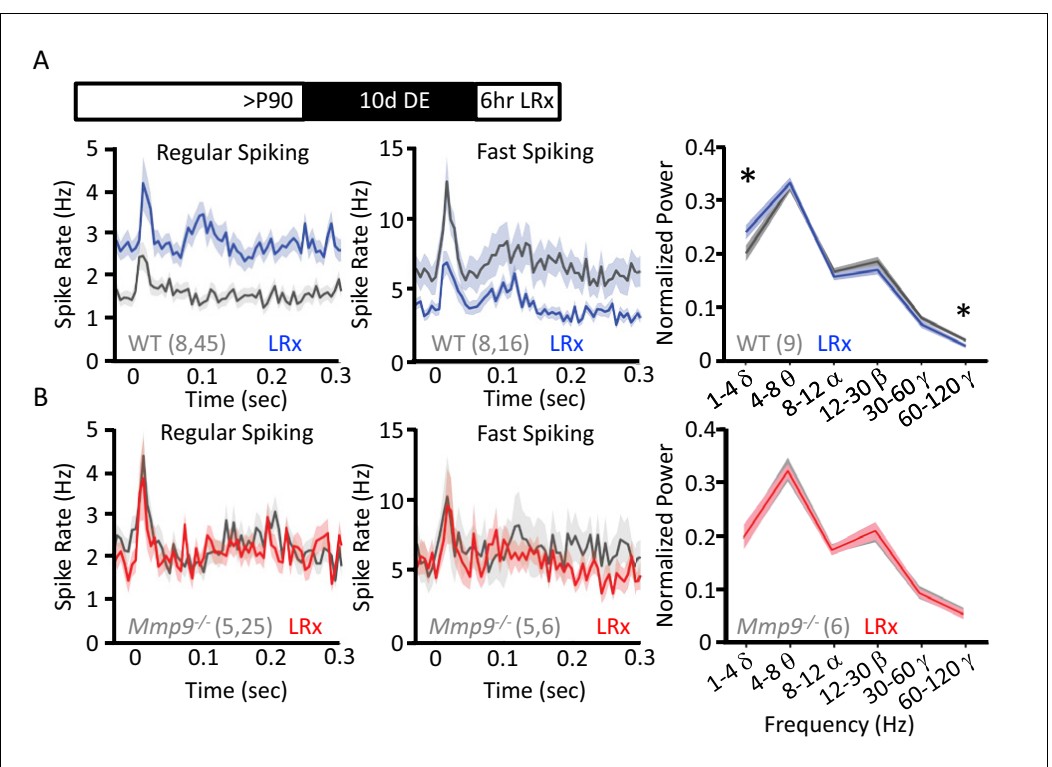

**Figure 5.** LRx regulates neuronal excitability and synchrony in WT but not *Mmp9$^{-/-}$* mice. (**A**) Inset: Experimental paradigm. 10 days of dark exposure, followed by LRx (6 hr) induced an increase in visually evoked activity of regular spiking neurons (RS; n=8 subjects, 45 units) and a decrease in evoked activity of fast spiking neurons (FS; n=8 subjects, 16 units) in WT mice. Post-stimulus time histograms (average ±SEM). 200 stimulus presentations of 100% contrast, 0.05 cycle per degree gratings, reversing at 1 Hz, stimulus onset = time 0 s. Right, Fourier transform of spontaneous LFP, normalized to within subject total power, and binned by frequency. In WT mice (n=9 subjects), LRx induces an increase in δ and a decrease in high γ power. (**B**) No change in regular spiking (n=5 subjects, 25 units) or fast-spiking (n=5 subjects, 6 units) visually evoked activity following LRx in *Mmp9$^{-/-}$* mice. Post-stimulus time histograms (average ±SEM). 200 stimulus presentations of 100% contrast, 0.05 cycle per degree gratings, reversing at 1 Hz, stimulus onset = time 0 s). Right, no change in oscillatory activity following LRx in *Mmp9$^{-/-}$* mice (n=6 subjects). *p<0.03; paired Student's T-test.

DOI: https://doi.org/10.7554/eLife.27345.013
The following source data and figure supplements are available for figure 5:

**Source data 1.**
DOI: https://doi.org/10.7554/eLife.27345.018
**Figure supplement 1.** Change in neuronal excitability by LRx in WTs is absent in *Mmp9$^{-/-}$* mice.
DOI: https://doi.org/10.7554/eLife.27345.014
**Figure supplement 1—source data 1.**
DOI: https://doi.org/10.7554/eLife.27345.016
**Figure supplement 2.** Normal functional organization of *Mmp9$^{-/-}$* mouse visual system.
DOI: https://doi.org/10.7554/eLife.27345.015
**Figure supplement 2—source data 1.**
DOI: https://doi.org/10.7554/eLife.27345.017

regulation (*Figure 5—figure supplement 2B*). Interestingly, visually-evoked single unit activity in *Mmp9*$^{-/-}$ mice was similar to that observed in WT adults after LRx (between subjects ANOVA, RS: F$_{(df,3,138)}$=6.792, p<0.001, Bonferroni *post hoc* WT+LRx v. *Mmp9*$^{-/-}$ p=1.00; FS: F$_{(df, 3,42)}$=0.045, Bonferroni *post hoc* WT+LRx v. *Mmp9*$^{-/-}$ p=1.00).

Despite the absence of a response to LRx, the structural and functional organization of the visual system of *Mmp9*$^{-/-}$ mice appeared normal. A current source density analysis revealed the typical distribution of current sinks and sources in *Mmp9*$^{-/-}$ mice, although the layer 5 current source was weaker and current sources and sinks were prolonged (*Figure 5—figure supplement 2A*). However, we observed no difference in the time to peak of the visually evoked field potential (VEP; evoked by 0.05 cpd 100% contrast gratings reversing at 1 Hz; *Figure 5—figure supplement 2B*). *Mmp9*$^{-/-}$ mice also retained a normal VEP contralateral bias, despite an increase in the amplitudes of both contralateral and ipsilateral eye VEPs (*Figure 5—figure supplement 2C*). Importantly, VEP amplitudes varied as a function of visual stimulus spatial frequency and contrast in both WT and *Mmp9*$^{-/-}$ mice (*Figure 5—figure supplement 2D*).

## LRx does not reactivate structural or functional ocular dominance plasticity in adult *Mmp9*$^{-/-}$ mice

To examine the impact of LRx on ocular dominance plasticity in *Mmp9*$^{-/-}$ mice, we examined the response to MD of the non-dominant, ipsilateral eye in post-critical period adults. MMP-9 activity has been primarily associated with the growth and strengthening of synapses (*Wang et al., 2008*; *Szepesi et al., 2014*). We therefore examined the impact of LRx on the strength of the non-deprived (i.e. dominant, contralateral) eye. In WT adults, LRx promoted a significant enhancement of non-deprived eye VEP following MD (between subjects repeated measures ANOVA, F$_{(df,1,8)}$=6.33, p=0.026, MD n=12, MD+LRx n=8). However, we did not observe an increase in the strength of the non-deprived input in *Mmp9*$^{-/-}$ mice, consistent with previous reports (*Spolidoro et al., 2012*; *Kelly et al., 2015*; *Pielecka-Fortuna et al., 2015*) and observed no reactivation of the response to MD after LRx (between subjects repeated measures ANOVA F$_{(df,1,5)}$=0.633, p=0.890; MD n=6, MD+LRx n=5; *Figure 6A*). During the critical period for ocular dominance plasticity, MD induces a significant decrease in the density of dendritic spines on pyramidal neurons in deprived V1b (contralateral to the occlusion), but this plasticity is lost with age (*Mataga et al., 2004*). Golgi staining of layer 4 neuronal dendrites reveals that LRx reactivated this form of structural plasticity in adult WT mice, and that MD following LRx induced a significant decrease in dendritic spine density in the deprived visual cortex (contralateral to the occluded eye; *Figure 6B*, one-way ANOVA, F=24.9, p<0.0001, p<0.05, Tukey-Kramer *post hoc*). However, we observed no change in dendritic spine density following MD or LRx+MD in *Mmp9*$^{-/-}$ mice.

## ECM degradation rescues structural and functional plasticity in *Mmp9*$^{-/-}$ mice

Our findings suggest that extracellular proteolysis, which is absent in *Mmp9*$^{-/-}$ mice, is central to the reactivation of structural and functional plasticity in the adult visual cortex by LRx. To test this hypothesis, we delivered an exogenous extracellular protease to the visual cortex of adult *Mmp9*$^{-/-}$ mice. Our initial attempt to rescue plasticity utilized active recombinant MMP-9. However, cortical delivery induced significant reactivity in astrocytes and microglia (robust staining for GFAP and Iba1, *Figure 3—figure supplement 1*). In contrast, delivery of hyaluronidase (Hyl), which degrades core components of the ECM, was well-tolerated (*Figure 3—figure supplement 1*). Importantly, hyaluronidase treatment induced a significant degradation of ECM labeling in *Mmp9*$^{-/-}$ mice, revealed by reductions in FITC-WFA and anti-aggrecan labelling (*Figure 7A*). The magnitude of the decrease in WFA labeling by hyaluronidase in *Mmp9*$^{-/-}$ mice was similar to that observed following LRx in WTs (*Figure 7B*). Hyaluronidase treatment did not induce a change in the excitability of FS INs (Baseline n=6 subjects, 9 units; following 7 days Hyl n=6 subjects, 14 units) or RS neurons (Baseline n=6 subjects, 19 units, following 7 days of Hyl n=9 subjects, 13 units; *Figure 7C*) or decrease the power of high-γ oscillations in *Mmp9*$^{-/-}$ mice. However, we observed a paradoxical increase in the power of high-γ (paired Student's T-test, p=0.028, n=6 subjects, *Figure 7D*). Importantly, hyaluronidase treatment was sufficient to rescue structural and functional plasticity in adult *Mmp9*$^{-/-}$ mice. Accordingly, MD induced an increase in the amplitude of the non-deprived (contralateral) eye VEP

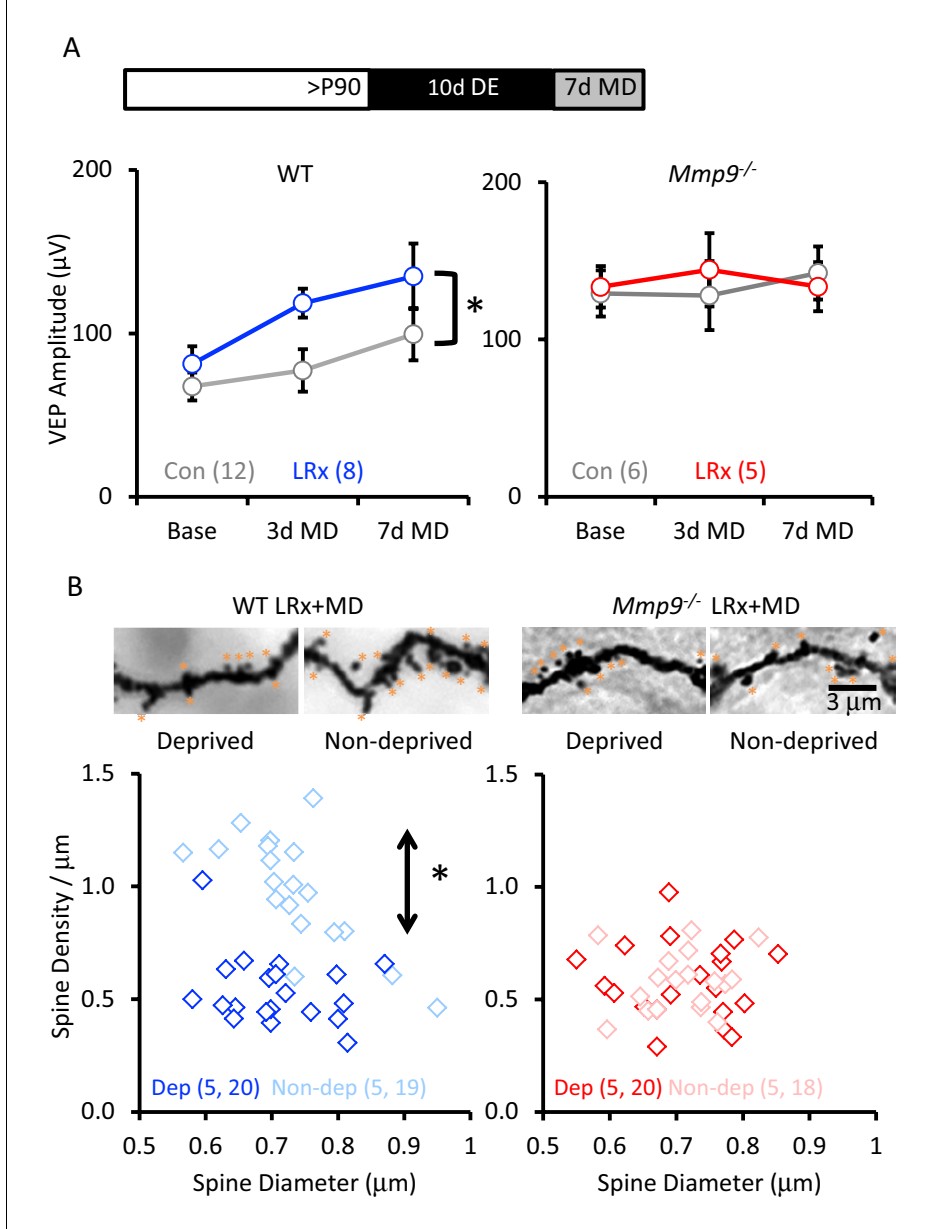

**Figure 6.** LRx reactivates structural and functional plasticity in visual cortex of adult WT, but not *Mmp9*[−/−] mice. (**A**) Inset: Experimental paradigm. VEP amplitudes in response to presentation of visual stimuli (100% contrast gratings, 0.05 cycle per degree, reversing at 1 Hz) to non-deprived contralateral eye, prior to monocular dperivation (MD) and following 3 and 7 days of MD. LRx promotes significant enhancement of non-deprived eye VEP following MD in WT adults (left, n=12, 8 subjects for Con and LRx, respectively) but not *Mmp9*[−/−] mice (right, n=6, 5 subjects for Con and LRx, respectively). Between subjects repeated measures ANOVA, WT: F=6.33, *p=0.026; *Mmp9*[−/−]: F=0.020, p=0.890. (**B**) Top: Representative images of Golgi stained basolateral dendrites of layer 4 neurons in binocular region of primary visual cortex following 7 days of MD in WT LRx (left) and *Mmp9*[−/−] LRx (right) subjects. Significant difference in the distribution of spine densities in deprived versus non-deprived V1b following 7 days of MD in WT LRx but not *Mmp9*[−/−] mice. Spine densities and diameters of the basolateral dendrites of layer 4 neurons were measured 75 – 100 µm from soma, spines are labeled with orange asterisks. One-way ANOVA, F=24.9, p<0.0001. KS tests (deprived versus non-deprived) WT LRx+MD Diameter: p=0.37; Density: p<0.0001; *Mmp9*[−/−] LRx+MD: Diameter: p=0.62; Density: p=0.95; n=(subjects, neurons).

DOI: https://doi.org/10.7554/eLife.27345.019

The following source data is available for figure 6:

**Source data 1.**

DOI: https://doi.org/10.7554/eLife.27345.020

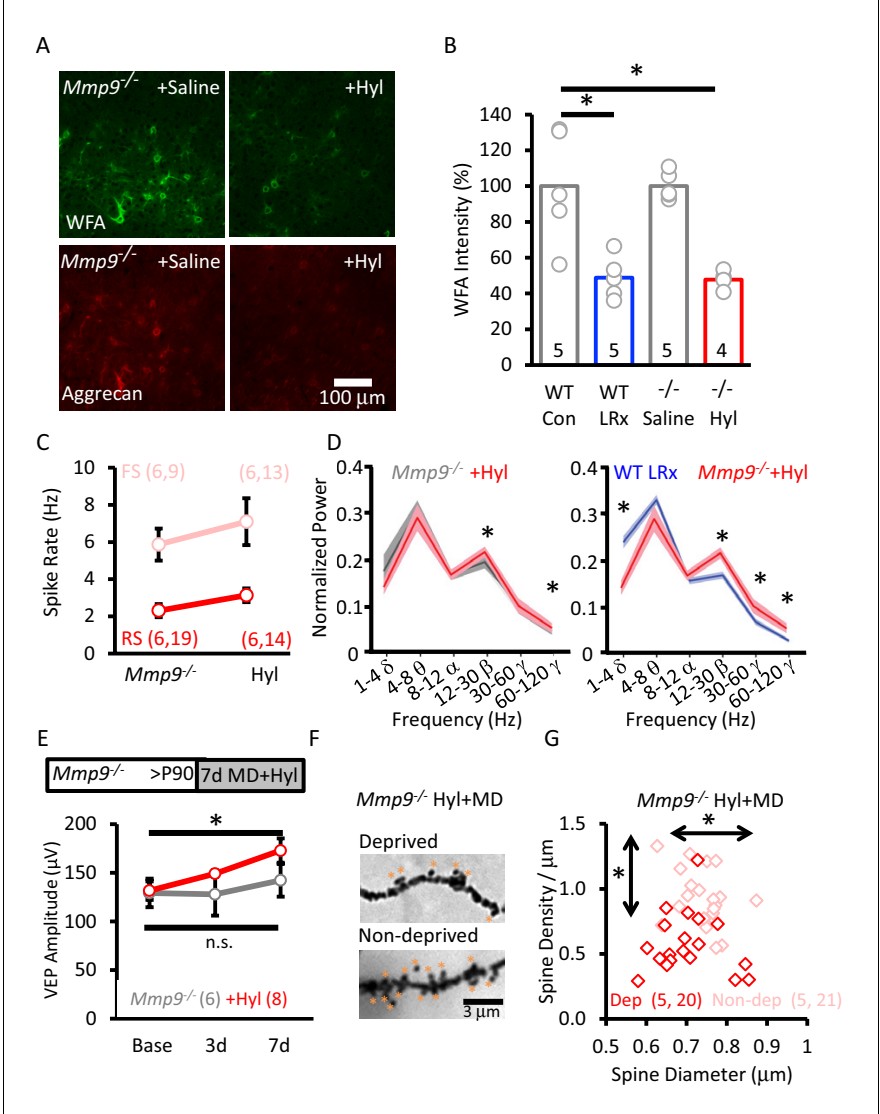

**Figure 7.** Rescue of structural and functional plasticity in *Mmp9*$^{-/-}$ mouse by hyaluronidase. (**A**) Hyaluronidase (Hyl, 4 µl of 200 U/ml at 100 nl/min) or vehicle was delivered i.c. via cannula at the initiation of LRx (1X/2 days for 7 days). Top: Reduction in WFA-FITC labeling following hyaluronidase treatment. Bottom: Reduction in aggrecan immunoreactivity following hyaluronidase treatment. (**B**) Reduction in WFA labelling following hyaluronidase treatment in *Mmp9*$^{-/-}$ mice compared to LRx in WT mice. WFA intensities was quantified 250 – 400 µm from cortical surface (WT Con: 100±14.3%, WT LRx: 48.8±5.3%, *Mmp9*$^{-/-}$ saline: 100.1±3.5%, *Mmp9*$^{-/-}$ Hyl: 47.8±2.6%, One-way ANOVA, F=12.7; p=0.0002, *p<0.01) n=subjects. (**C**) No change in the excitability of FS neurons from *Mmp9*$^{-/-}$ mice following hyaluronidase treatment (light red, n=6 subjects, 9 units; +Hyl n=6 subjects, 13 units) or RS (red, n=6 subjects, 19 units; +Hyl n=6 subjects, 14 units). (**D**) Hyluronidase-induced change in neuronal synchrony does not mimic the response to LRx in WTs (n=6 subjects, *p<0.03, paired Student's T-test). Comparison of oscillatory activity in two cohorts of subjects that exhibit robust adult ocular dominance plasticity: WT+LRx (n=9 subjects) and *Mmp9*$^{-/-}$+Hyl; *p<0.03, unpaired Student's T-test. (**E**) Inset: Experimental paradigm. Hyaluronidase treatment (n=8 subjects; red) promotes significant enhancement of non-deprived eye VEP following monocular deprivation in adult *Mmp9*$^{-/-}$ mice (n=6 subjects; gray). Repeated measures ANOVA, F=16.668, p=0.001, Bonferroni *post-hoc* *p=0.001, Baseline versus 7d. MD alone, repeated measures ANOVA, F=0.446, p=0.655 (n.s.=not significant). (**F**) Representative images of Golgi stained basolateral dendrites of layer 4 neurons in V1bfollowing 7 days of MD in *Mmp9*$^{-/-}$+Hyl subject. (**G**) Significant difference in distribution of spine densities and diameters in deprived versus non-deprived visual cortex following 7 days of MD in *Mmp9*$^{-/-}$+Hyl mice, quantified 75 – 100 µm from soma). One-way ANOVA, F=6.6, p<0.0001. KS for diameter: p=0.0149; density: p<0.0001. n=(subjects, neurons).

*Figure 7 continued on next page*

*Figure 7 continued*

DOI: https://doi.org/10.7554/eLife.27345.021

The following source data is available for figure 7:

**Source data 1.**

DOI: https://doi.org/10.7554/eLife.27345.022

(Hyl+MD, repeated measures ANOVA, $F_{(df,2,8)}$=16.668, p=0.001; Bonferroni *post-hoc* p=0.001, Base v. 7d; MD alone, repeated measures ANOVA, $F_{(df,2,6)}$=0.446, p=0.655, n.s. *Figure 7E*) and an increase in the density and diameter of pyramidal neuron dendritic spines in non-deprived V1b (ipsilateral to the occlusion; *Figure 7F and G*).

## Discussion

The utility of binocular visual deprivation to reactivate the critical period for ocular dominance plasticity has been repeatedly demonstrated (*He et al., 2006*) (*He et al., 2007*; *Montey and Quinlan, 2011*) (*Duffy and Mitchell, 2013*; *Stodieck et al., 2014*; *Eaton et al., 2016*; *Mitchell et al., 2016*). However, little is known regarding the mechanism by which dark exposure promotes synaptic plasticity. Here, we demonstrate that light reintroduction (LRx) following dark exposure (DE) induces an increase in the perisynaptic activity of matrix metalloproteinase 9 (MMP-9). Importantly, the increase in MMP-9 activity by LRx occurs at thalamic inputs to the cortex, identifying this class of synapses as central to the reactivation of plasticity in adults. Furthermore, structural and functional ocular dominance plasticity, which are absent in $Mmp9^{-/-}$ mice, are rescued by hyaluronidase, an enzyme that degrades core components of the ECM. Thus, LRx represents a minimally invasive method to recruit proteolytic pathways that reverse developmental constraints on structural and functional plasticity in adult primary visual cortex.

### LRx reactivates plasticity in adult visual cortex

Previous studies employing DE have assumed that the elimination of visual input was sufficient to reactivate plasticity in the adult visual cortex. Indeed, biochemical and slice electrophysiology experiments performed on subjects sacrificed in the dark reveal changes in cortical circuitry predicted to lower the threshold for synaptic plasticity (*Cooper and Bear, 2012*). For example, DE returns the composition of the NMDA subtype of glutamate receptor to the 'juvenile' form (containing high levels of the NR2B subunit), and enhances the temporal summation of NMDAR-mediated currents (*He et al., 2006*; *Yashiro et al., 2005*). In addition, following DE, forms of synaptic plasticity typically limited to juveniles are re-expressed (*Huang et al., 2010*), the excitability of pyramidal neurons is enhanced, and the integration window for spike-timing dependent plasticity is increased (*Goel and Lee, 2007*; *Guo et al., 2012*). Similarly, brief inactivation of retinal output by intraocular TTX injection promotes the reversal of anatomical and physiological responses to MD in mice and kittens (*Fong et al., 2016*). Interestingly, the increase in pyramidal neuron excitability following DE is associated with an increase in the strength of cortico-cortical but not thalamo-cortical synapses (*Petrus et al., 2015*). However, DE alone does not induce an increase in MMP-9 activity or extracellular proeolysis at thalmo-cortical synapses. In contrast, we find that LRx following DE increases perisynaptic MMP-9 activity which is required to reactivate robust structural and functional plasticity in post-critical period adults. Accordingly, the reactivation of plastcity previously attributed to DE alone was likely induced by the LRx necessary to perform in vivo physiological and psychophysical measurements of visual function (*Gu et al., 2013*; *Gu et al., 2016*; *Montey and Quinlan, 2011*; *Eaton et al., 2016*; *Duffy and Mitchell, 2013*; *Stodieck et al., 2014*).

DE and LRx are relatively non-invasive, and therefore make attractive candidates for translation to clinical treatment of amblyopia. However, the reactivation of plasticity by LRx is transient, and must be quickly harnessed by instructive visual experience to promote recovery of amblyopia in rodents (*Eaton et al., 2016*). The time course of the plasticity induced by LRx may be determined by regrowth of the ECM, which mirrors the reformation of the mature ECM observed after enzymatic degradation (*Lensjø et al., 2017*; *Romberg et al., 2013*). In addition, the sensitivity to LRx is developmentally regulated and includes a refractory period in late adolescence (*Huang et al., 2010*)

which may explain the modest response to DE at some ages (*Erchova et al., 2017*). We propose that developmental changes in the dynamics of ECM synthesis/degradation may also underlie the expression of the refractory period for reactivation of plasticity by LRx.

## Extracellular proteolysis and the regulation of neuronal excitability

The majority of work correlating extracellular proteolysis with enhanced synaptic plasticity claims to focus on the consequence of degrading the dense PNNs that enshroud PV[+] INs. However, the exogenous lyases typically used to degrade PNNs, including hyaluronidase and chondroitinase, have multiple, overlapping substrates within PNNs and within diffuse ECM (i.e. hyaluronin and chondroitin sulfates; *Tipton, 1992*). Similarly, following LRx we observe an increase in MMP activity and a decrease in ECM density in PV[+] and PV[-] regions that co-localize with a marker for thalamic axons, suggesting degradation of ECM at synapses onto PNN- and non-PNN-bearing neurons. The LRx-induced increase in MMP-9 activity at synapses onto PV[+] INs predicted to reduce the strength of these synapses, and enhance pyramidal neuron excitability via disinhibition. We hypothesize that subsequent activity-dependent change in the strength of excitatory inputs onto RS neurons are enabled by baseline MMP-9 activity. Consequently, MMP-9 proteolysis is predicted to promote weakening of excitatory synapses onto inhibitory neurons and promote strengthening of excitatory synapses onto excitatory neurons. Degradation of the extracellular environment has previously been shown to exert different effects at different classes of synapses. For example, application of chondroitinase ABC (chABC) to PNN[+] FS principal neurons in the auditory brainstem (MNTB) and PV[+] INs in the cortex decreases neuronal excitability and gain (*Balmer, 2016*). In contrast, in dissociated hippocampal interneurons, chABC reduces the current threshold and after-hyperpolarization, indicative of increased excitability (*Dityatev et al., 2007*). However, chABC enables synaptic plasticity without changing synaptic gain or excitability of PNN[+] non-fast-spiking pyramidal neurons of CA2 region of the hippocampus (*Carstens et al., 2016*). Thus, the relationship between ECM/PNN integrity and neuronal excitability is complex and differs across brain regions and in PNN bearing versus non-PNN bearing neurons.

The weakening of thalamic inputs to FS INs is a novel role for MMP-9, which is typically associated with the growth and strengthening of excitatory synapses onto excitatory neurons (*Kaliszewska et al., 2012*; *Lebida and Mozrzymas, 2016*; *Michaluk et al., 2011*; *Ganguly et al., 2013*), but see (*Romberg et al., 2013*). Indeed, inhibition of MMP-9 blocks the maintenance of long-term potentiation (LTP) (*Wiera et al., 2013*; *Nagy et al., 2006*), and the enlargement of pyramidal neuron dendritic spines following tetanic stimulation of hippocampal CA1-CA3 synapses (*Wang et al., 2008*; *Szepesi et al., 2014*). Both high frequency stimulation and a chemical LTP protocol have been shown to increase the activity of MMP-9, and may preferentially promote the enlargement of small spines (*Bozdagi et al., 2007*; *Szepesi et al., 2014*). Inhibition of MMP-9 also blocks many forms of learning, including water maze navigation (*Meighan et al., 2006*), fear conditioning (*Nagy et al., 2007*), addiction (*Brown et al., 2007*) drug withdrawal and relapse (*Smith et al., 2014*). Pharmacological inhibition and genetic deletion of *Mmp9* hvebeen shown to impair ocular dominance plasticity during the critical period (*Kelly et al., 2015*) and block the strengthening of non-deprived eye responses (*Spolidoro et al., 2012*; *Pielecka-Fortuna et al., 2015*). Similarly, the strengthening of the response of the non-deprived eye observed following prolonged MD in adults, which we show to be significantly enhanced by LRx in WT adults, is absent in *Mmp*9[−/−] mice.

The differential distribution of ECM molecules, including the concentration into PNNs, appears normal in *Mmp*9[−/−] mice. However, RS neurons in *Mmp*9[−/−] visual cortex have elevated excitability and reduced dendritic spine density, as previously reported (*Murase et al., 2016*; *Kelly et al., 2015*). Importantly, the increase in neuronal excitability in *Mmp*9[−/−] mice was reflected in visually-evoked but not spontaneous activity of RS neurons, reinforcing the idea that feed-forward thalamic inputs to cortex are important targets for regulation by MMP-9. Interestingly, hyaluronidase rescued plasticity in adult *Mmp*9[−/−] mice without inducing a change in neuronal excitability. However, the enhanced baseline excitability of RS neurons in *Mmp*9[−/−] mice may preclude the need to further enhance excitability to reactivate plasticity. Nonetheless, the rescue of structural and functional plasticity in *Mmp*9[−/−] mouse by hyaluronidase supports the assertion that degradation of the extracellular environment reactivates ocular dominance plasticity in adults (*Pizzorusso et al., 2002*; *Pizzorusso et al., 2006*).

## Targeted proteolysis by MMP-9

MMP zymography has previously been applied to polyacrylamide gels, tissue slices and low-density neuronal culture (*Szklarczyk et al., 2002*; *Wilczynski et al., 2008*; *Szepesi et al., 2014*), and is emerging as a powerful tool to report extracellular protolysis in vivo (*Bozdagi et al., 2007*; *Smith et al., 2014*). Consistent with previous studies in the hippocampus (*Wilczynski et al., 2008*; *Bozdagi et al., 2007*; *Gawlak et al., 2009*), we demonstrated that MMP activity co-localizes with excitatory synapses. In addition, we show that the increase in perisynaptic MMP activity induced by salient experience is associated with a specific class of synapse (thalamic inputs to cortex), and a decrease in aggrecan, a known MMP-9 substrate (*Mercuri et al., 2000*). Hyaluronidase treatment of cultured neurons also reduces immunoreactivity for aggrecan (*Giamanco et al., 2010*), suggesting the two enzymes may share common substrates. However degradation of hyaluronic acid, the primary target of hyaluronidase, would also reduce aggrecan clustering. Importantly, targets of both lyases are found in throughout the extracellular environment (*Verslegers et al., 2013*). Interestingly, hyaluronidase regulates the lateral mobility of AMPARs on spiny pyramidal neurons but not aspiny interneurons, suggesting different targets at different types of synapses (*Frischknecht et al., 2009*; *Klueva et al., 2014*). In addition to ECM molecules, MMP-9 proteolytically processes a variety of growth factors (*Yu and Stamenkovic, 2000*; *Mizoguchi et al., 2011*), cell surface glycoproteins (*Michaluk et al., 2007*) and cell adhesion molecules/receptors (*Bajor et al., 2012*; *Peixoto et al., 2012*). The regulation of MMP activity occurs at many levels, including transcription, translation, release, cleavage and activation of the pro-enzyme. Accordingly, a serine protease, such as tissue plasminogen activator, can cleave and activate MMPs (*Oray et al., 2004*; *Mataga et al., 2004*). MMP-9 activity is also controlled by endogenous inhibitors such as tissue inhibitor of matrix metalloproteinase (TIMP; *Candelario-Jalil et al., 2009*). One interesting possibility is that MMP-9 and TIMP activity are both reduced by DE, but MMP-9 activity is activated more quickly than TIMP activity following LRx, resulting in a transient increase in perisynaptic proteolysis. This work underscores the importance of bidirectional control of synaptic plasticity by manipulations of extracellular signaling pathways, and highlights the therapeutic potential of extracellular proteolysis for the recovery of structural and functional plasticity of adult circuits.

# Materials and methods

## Subjects

C57BL/6J and *Mmp9*$^{-/-}$ (007084, B6 background) mice were purchased from Jackson Laboratory (Bar Harbor, ME). Equal numbers of adult (>postnatal day 90, >P90) males and females were used. Animals were raised in 12/12 hr light/dark cycle unless specified. All procedures conformed to the guidelines of the University of Maryland Institutional Animal Care and Use Committee. Experiments were performed (or subjects were sacrificed) 6 hr into the light phase of a 12/12 hr cycle.

## Monocular deprivation

Subjects were anesthetized with ketamine/xylazine (100 mg/ 10 mg/kg, i.p.). The margins of the upper and lower lids of one eye were trimmed and sutured together using a 5–0 suture kit with polyglycolic acid (CP Medical). Subjects were returned to their home cage after recovery at 37°C for 1–2 hrs, and disqualified in the event of suture opening.

## Reagents

CSPGs were visualized with 5 μg/ml fluorescein wisteria floribunda lectin (WFA, Vector Labs). MMP-9 Inhibitor 1 (EMD) was dissolved in dimethyl sulphoxide (DMSO) to 2 mM, then diluted to 5 μM with saline and sterilized with a 0.2 μmφ filter to make a 1000X stock. Final concentration (5 nM) diluted in sterile saline. MMP biomarker (dye-quenched gelatin; D12054, ThermoFisher Scientific) was diluted to 2 mg/ml with saline and sterilized with a 0.2 μmφ filter. Recombinant mouse MMP-9 (rmMMP-9, R & D Systems) was activated following vendor instructions and diluted to 1 μg/ml with sterile saline. Hyl was diluted to 200 U/ml with saline and sterilized with a 0.2 μmφ filter.

## Antibodies

The following antibodies/dilutions were used: mouse anti-parvalbumin (PV) (Millipore) RRID:AB_2174013, 1:2000; rabbit anti-aggrecan (Millipore) RRID:AB_90460, 1:500; rabbit anti-MMP-9 (Cell Signaling) RRID:AB_2144612, mouse anti-β-actin (Sigma-Aldrich) RRID:AB_476744, guinea pig anti-VGluT1 and anti-VGluT2 (Millipore) RRID:AB_2301751 and RRID:AB_1587626, 1:2000; rabbit anti-GFAP (DAKO) RRID:AB_10013382, 1:2000; rabbit anti-Iba-1 (Wako Chemicals USA) RRID:AB_839504, 1:500; followed by appropriate secondary antibodies: goat anti-mouse, anti-rabbit and anti-guinea pig IgG conjugated to Alexa-488, 546 or 647 (Life Technologies) RRID:AB_2534089, RRID:AB_2534093, RRID:AB_2535805, RRID:AB_2534118, 1:1000.

## In vivo delivery of reagents

A micro-osmotic pump 1007D (Alzet) was used to infuse MMP-9 inhibitor, beginning 6 days prior to the initiation of LRx, concurrent with DE. The pumps were filled with 100 μl saline or 5 nM MMP-9 inhibitor and connected to brain infusion kit 3 (Alzet) with a cannula (2 mm projection) implanted 500 μm medial and dorsal to V1b. Other reagents were delivered through cannulae (2 mm projection, PlasticsOne) implanted ~3 weeks prior to injection. A total volume of 4 μl at 100 nl/min was delivered via a Hamilton syringe attached to a Microsyringe Pump Controller (World Precision Instruments). MMP biomarker (2 mg/ml) solution was delivered at the onset of LRx, i.e. 24 hr prior to perfusion. Hyaluronidase (Hyl, Sigma-Aldrich, H-1136, 200 U/ml) was delivered once every 2 days for 7 days.

## Immunohistochemistry

Subjects were perfused with phosphate buffered saline (PBS) followed by 4% paraformaldehyde (PFA) in PBS. Brain was post-fixed in 4% PFA for 24 hr followed by 30% sucrose for 24 hr, and cryo-protectant solution (0.58 M sucrose, 30% (v/v) ethylene glycol, 3 mM sodium azide, 0.64 M sodium phosphate, pH 7.4) for 24 hr prior to sectioning. Coronal sections (40 μm) were made on a Leica freezing microtome (Model SM 2000R). Sections were blocked with 4% normal goat serum (NGS) in 1X PBS for 1 hr. Antibodies were presented in blocking solution for 18 hr, followed by appropriate secondary antibodies.

## Confocal imaging and analysis

Images were acquired on a Zeiss LSM 710 confocal microscope. The cortical distribution of ECM molecules and PV immunoreactivity was examined in a z-stack (3 × 7.5 μm images) acquired with a 10X lens (Zeiss Plan-neofluar 10x/0.30, NA=0.30). Maximal intensity projections (MIPs; 450 μm width, 750 μm depth from cortical surface) were used to obtain mean intensity profiles in FIJI (NIH). For WFA staining, a MIP of a z-stack (11 slices x 0.5 μm images) was acquired at 40X (Zeiss Plan-neofluar 40x/1.3 Oil DIC, NA=1.3) and 100X (Zeiss Plan-neofluar 100x/1.3 Oil DIC, NA=1.3). WFA$^+$ cells were selected based on size threshold (>36 μm$^2$) and fluorescence threshold determined using a histogram-derived global thresholding method based on IsoData algorithm in FIJI. The same threshold was used for all images. In *Figure 1*: PV$^+$ somata were identified by size exclusion (80 – 300 μm$^2$) and fluorescence intensity thresholding. Region of interest (256 pixels) was defined by a line scan (single pixel height) through the center of a PV$^+$ soma (x, y)=(a, b), WFA intensity was determined for PV$^+$ and PV$^-$ pixels. In *Figure 3*: Co-localization of MMP biomarker puncta with VGluTs and PV was analyzed in a single Z-section image taken at 40X, using the JACoP plugin "object-based method" in Fiji. Co-localization was based on centers of mass particle coincidence (size threshold >0.15 μm$^2$) after fluorescence intensity thresholding.

## Western blot analysis

Mice were anesthetized with isoflurane vapor (4% in 100% $O_2$) and sacrificed following cervical dislocation. The primary visual cortex and medial prefrontal cortex were rapidly dissected in ice-cold dissection buffer (212.7 mM sucrose, 2.6 mM KCl, 1.23 mM $NaH_2PO_4$, 26 mM $NaHCO_3$, 10 mM dextrose, 1.0 mM $MgCl_2$, 0.5 mM $CaCl_2$, 100 μM kynurenic acid) saturated with 95% $O_2$/5% $CO_2$. V1b was isolated using the lateral ventricle and dorsal hippocampal commissure as landmarks. Tissue was homogenized using a Sonic Dismembrator (Model 100, Fisher Scientific) in ice-cold lysis buffer (150 mM NaCl, 1% Nonidet P-40, 50 mM Tris-HCl, pH8.0) containing a protease inhibitor cocktail

(Cat#11697498001, Roche). Protein concentration of the homogenate was determined using the BCA Protein Assay kit (Pierce). Equal amounts of total protein (20 μg per lane) were applied to a 12% SDS-polyacrylamide gel for electrophoresis, then transferred to a nitrocellulose membrane. The membranes were incubated with a blocking solution (4% skim milk in 1X PBS) for 30 min. Antibodies were presented in the blocking solution for 2 hr, followed by appropriate secondary antibodies. Immunoreactive bands were visualized with a Typhoon TRIO Variable Imager (GE Healthcare). The intensity of immunoreactive bands of active MMP-9 (95 kDa), which can be distinguished from pro-MMP-9 (~105 kDa) (*Szklarczyk et al., 2002*), was analyzed using ImageQuant TL (rubber band background subtraction; GE Healthcare) and normalized to β-actin (38 kDa).

## Golgi staining and dendritic spine density analysis

Golgi staining was produced with FD Rapid GolgiStain Kit (FD Neuro Technologies) as per manufacturer's instructions. The brains were immersed in solution A+B for 7 days at room temperature, and transferred to solution C for 3 days at 4℃. Coronal sections (100 μm) were made with a Leica VT100S vibrating microtome and mounted on gelatin-coated slides (FD Neuro Technologies). Neurolucida (MBF Bioscience, St. Albans, VT) with an Olympus BX61 light microscope was used to trace dendritic morphologies. Using a 40X lens (Olympus Plan N, NA=0.65), dendritic arbors of layer 4 neurons were traced, followed by sholl analysis in Neuroexplorer (MBF Bioscience) to identify the region 75 to 100 μm from soma of basolateral dendrites, which were then traced using a 100X lens (Olympus Plan N Oil, NA=1.25). Dendritic protrusions lengths at least 50% greater than the dendritic shaft diameter were identified as spines, and counted in Neuroexplorer.

## Chronic in vivo recordings

Adult mice were anesthetized with 2.5% isoflurane in 100% $O_2$ and a handmade 1.2 mm 16-channel shank electrode was implanted in V1b (stereotaxic coordinates from Bregma: anterior/posterior, 2.8 mm; medial/lateral, 3.0 mm; dorsal/ventral, 1.2 mm). After recovery of righting reflex animals were administered buprenorphine (0.1 mg/kg, i.p.) for post-surgical analgesia and returned to their home cage. Awake, head fixed recordings began 5–7 days after implantation. Activity was evoked by passive viewing of 200 × 1 s trials of square-wave gratings (0.05 cycles per degree (cpd), 100% contrast, reversing at 1 Hz, via MATLAB with Psychtoolbox extensions, (*Brainard, 1997*; *Pelli, 1997*)). Spontaneous activity was recorded during passive viewing of a grey screen of equal luminance (26 cd/m$^2$). VEPs evoked with high contrast, low spatial frequency visual stimuli are larger in $Mmp9^{-/-}$ mice than controls, therefore, the effect of visual stimulus spatial frequency on VEP amplitudes was examined at 50% contrast

Multiunit activity (isolated with 300 Hz high pass and 5 kHz low pass filters) and local field potentials (isolated with 300 Hz low pass filter and 60 Hz notch-filter) were collected using a RZ5 bioamp processor and RA16PA preamp (Tucker Davis Technologies, TDT). Multiunit activity was sorted into single units using a Bayesian clustering algorithm in OpenSorter (TDT) and further analyzed using custom MATLAB routines (*Lantz, 2017*; copy archived at https://github.com/elifesciences-publications/InVivo). Sorted single units were classified as fast-spiking (presumptive PV$^+$ INs) or regular spiking (presumptive pyramidal neurons) based on three parameters: slope of the waveform 0.5 msec after the trough, time elapsed between the trough and peak, and the ratio of trough to peak height (*Niell and Stryker, 2008*). Units were assigned to cortical layers based on VEP waveform, and units assigned to layers 2/3 and 4 were combined and included in the analysis. Visually-evoked local field potentials (VEPs) assigned to layer 4 were averaged over 200 trials using custom MATLAB routines (*Lantz, 2017*). Power spectra from baseline LFP were calculated using the Fast Fourier Transform in MATLAB paired with Chronux extensions. Power spectra were normalized to total power for each subject and frequency bands were defined as: δ (1–4 Hz), θ (4–8 Hz), α (8–12 Hz), β (12–30 Hz), γ (30–60 Hz) and high- γ (60–120 Hz).

## Statistics

An unpaired two-tailed Student's T-test was used to determine the significance between two independent experimental groups, and a paired Student's T-test was used for two measurements within the same subject. A one-way ANOVA was used to determine the significance between three or more independent experimental groups. Repeated measures ANOVA was used to determine the

significance with more than two measures within the same subjects followed by a Tukey-Kramer honestly significant difference post hoc (ex vivo data) or Bonferroni post hoc (in vivo data) for pairwise comparisons, when ANOVA was p<0.05 (SPSS, IBM). A Kolmogorov-Smirnov test was used to determine the significance between the distributions of two independent data sets. Multi-dimensional Kolmogorov-Smirnov test was used for data sets with two independent measurements (MATLAB).

## Additional information

### Funding

| Funder | Grant reference number | Author |
|---|---|---|
| National Eye Institute | R01 | Elizabeth Quinlan |

The NEI had no role in study design, data collection and interpretation, or the decision to submit the work for publication.

### Author contributions
Sachiko Murase, Crystal L Lantz, Data curation, Formal analysis, Investigation, Methodology; Elizabeth M Quinlan, Conceptualization, Data curation, Formal analysis, Funding acquisition, Investigation, Methodology, Project administration

### Author ORCIDs
Sachiko Murase (iD) http://orcid.org/0000-0002-9078-0471
Crystal L Lantz (iD) http://orcid.org/0000-0002-9763-4725
Elizabeth M Quinlan (iD) http://orcid.org/0000-0003-3496-6607

### Ethics
Animal experimentation: All procedures, under Quinlan lab protocol R-16-30, conformed to the guidelines of the University of Maryland Institutional Animal Care and Use Committee and the Guide for the Care and Use of Laboratory Animals of the National Institutes of Health.

### Decision letter and Author response
Decision letter https://doi.org/10.7554/eLife.27345.024
Author response https://doi.org/10.7554/eLife.27345.025

## Additional files

### Supplementary files
• Transparent reporting form
DOI: https://doi.org/10.7554/eLife.27345.023

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
