## [Decision Letter]

Thank you for submitting your article "Light reintroduction after dark exposure reactivates plasticity in adults via perisynaptic activation of MMP-9" for consideration by *eLife*. Your article has been reviewed by three peer reviewers, one of whom, Sacha B Nelson (Reviewer #1), is a member of our Board of Reviewing Editors, and the evaluation has been overseen by Andrew King as the Senior Editor. The following individual involved in review of your submission has agreed to reveal their identity: Takao K Hensch (Reviewer #3).

The reviewers have discussed the reviews with one another and the Reviewing Editor has drafted this decision to help you prepare a revised submission.

Summary:

Although it has been known for some time that the critical period for visual cortical plasticity can be reactivated by dark rearing, less is known about the underlying mechanisms. In the present study the authors show convincingly that it is not dark-rearing itself, but the re-exposure to light that triggers reactivation of plasticity, and that this occurs through the action of the MMP9 protease on the extracellular matrix. Although this enzymatic activity had previously been implicated in visual plasticity, the present results link these mechanisms to the effects of altered visual experience following dark rearing. The finding that the enzyme hyaluronidase is sufficient to rescue structural and functional plasticity has wide-reaching implications for therapeutic strategies.

Essential revisions:

1) Reviewers agreed that a limitation of the approach is the lack of specificity using the potentially off-target enzyme hyaluronidase to rescue the MMP-9 phenotype. One reviewer suggested that other MMP inhibitors or MMP KO mice (showing no effect) could be used to further confirm the specificity. Upon discussing this issue, the reviewers and editors agreed that textual revisions clearly indicating potential differences in the substrates engaged by the different manipulations, as well as a more careful consideration of the context of prior molecular studies using this paradigm would suffice.

2) Reviewers found the micrographs illustrating Golgi stained dendrites unconvincing and the accompanying spine quantification insufficiently described and documented. There were concerns that the methods may have resulted in surprisingly large changes in part as a function of the precise sampling used (some further details relating to this concern are given below).

3) There is a long list of minor edits indicating many details of the presentation that need clarification or correction.

Reviewer #3:

Figure 1: label for the panel 'D' is missing from figure. In addition, the data presentation in this panel is not intuitive. Does each data point represent a cell (presumably WFA+)? What are the numbers on each axis (off PV and on PV) represent? How does one WFA+ cell appear to be off PV and on PV at the same time (but to different extent)? The message of this panel is 'the decrease in WFA staining following LRx was observed on and off PV+ cell bodies' (Results paragraph one) but this graph fails to show this right away. Furthermore, does LRx preferentially affect WFA intensity in on PV or off PV population? Please also state how many animals were used in this analysis apart from total cell numbers.

Figure 3, panel C: it is unclear how the quantification was carried out. In the sample micrograph, two arrows (blue and red) are present but they are not explained in the figure legend. Do they represent 'on PV' and 'off PV' populations of VGluT2 respectively? What is the criteria for classifying 'on PV' or 'off PV' VGlut2 puncta? The use of same colour scheme (red and blue) for Con and LRx groups in quantification below also becomes confusing. Two micrographs (at high magnification) should be included to compare between Con and LRx conditions. Also, does LRx increase co-localization between MMP biomarker and VGluT2 in both on PV/off PV populations? The figure legend suggests a 'parallel increase' (Figure 3 legend) but the quantification graphs otherwise show a significant difference in 'off PV' only. The n numbers for this panel are questionable – does the data come from '12, 8 cells' or '6, 7 slices for Con and LRx'?

Figure 6, panel A: why does MMP-9 KO mice non-deprived eye have very high VEP amplitude (>100uV) at base level (before MD) even in Con (no DE) condition? Given Kelly et al., 2015 show that MMP-9 KO mice have normal ocular dominance levels similar to that of WT mice, this would suggest that MMP-9 KO mice have even higher contralateral response (perhaps >200uV). This contradicts Figure 5—figure supplement 2 which shows much lower VEP responses from both contralateral and ipsilateral eyes of MMP-9 KO mice. In addition, Kelly et al. showed that MMP-9 KO mice showed an OD shift after 7d MD (increase in non-deprived eye response), which does not match with the data presented here. Please explain these discrepancies.

Figure 6, panel B: Sample images of Golgi stained basolateral dendrites are very blurry and it is hard to appreciate how individual spines can be identified for quantification. Better images with clear labeling of identified spines would be necessary.

[Editors' note: further revisions were requested prior to acceptance, as described below.]

Thank you for submitting your article "Light reintroduction after dark exposure reactivates plasticity in adults via perisynaptic activation of MMP-9" for consideration by *eLife*. Your article has been reviewed by three peer reviewers, one of whom, Sacha B Nelson (Reviewer #1), is a member of our Board of Reviewing Editors, and the evaluation has been overseen by Andrew King as the Senior Editor. The following individuals involved in review of your submission have agreed to reveal their identity: Serena Dudek (Reviewer #2); Takao K Hensch (Reviewer #3).

The reviewers have discussed the reviews with one another and the Reviewing Editor has drafted this decision to help you prepare a revised submission.

Please address the remaining points. This should not need to be sent back to reviewers. The manuscript may be considered "accepted pending minor revisions."

Reviewer #1:

I have no further concerns

Reviewer #2:

The authors have adequately addressed my concerns, and in my opinion, those of the other reviewers, in this revision. It is an important contribution to the literature.

Reviewer #3:

Hyaluronidase non-specificity is the major limitation of this study. But since it can rescue MMP-9 KO phenotype, some common substrates or pathways are suggested to be targeted by these two enzymes. The authors should focus their discussion on potentially shared downstream targets.

---

## [Author Response]

Essential revisions:1) Reviewers agreed that a limitation of the approach is the lack of specificity using the potentially off-target enzyme hyaluronidase to rescue the MMP-9 phenotype. One reviewer suggested that other MMP inhibitors or MMP KO mice (showing no effect) could be used to further confirm the specificity. Upon discussing this issue, the reviewers and editors agreed that textual revisions clearly indicating potential differences in the substrates engaged by the different manipulations, as well as a more careful consideration of the context of prior molecular studies using this paradigm would suffice.

We have added the following to the Discussion:

“Although hyaluronidase rescues structural and functional plasticity in Mmp9^-/-^ mouse, it is not yet known if these two enzymes target the same substrates. Hyaluronic acid, the primary target of hyaluronidase, is found in diffuse and condensed ECM. However, hyaluronidase regulates the lateral mobility of AMPARs on spiny pyramidal neurons but not aspiny interneurons, suggesting different targets at different classes of synapses (Frischknecht et al., 2009)(Klueva, Gundelfinger, Frischknecht and Heine, 2014). MMP-9 also has targets in diffuse and condensed ECM (Verslegers, Lemmens, Van Hove and Moons, 2013) and proteolytically processes a variety of growth factors (Yu and Stamenkovic, 2000)(Mizoguchi et al., 2011), cell surface glycoproteins (Michaluk et al., 2007) and cell adhesion molecules / receptors (Bajor et al., 2012)(Peixoto et al., 2012). The regulation of MMP activity occurs at many levels, including transcription, translation, release, cleavage and activation of the pro-enzyme. Accordingly, a serine protease, such as tissue plasminogen activator, can cleave and activate MMPs (Oray et al., 2004)(Mataga et al., 2004).”

2) Reviewers found the micrographs illustrating Golgi stained dendrites unconvincing and the accompanying spine quantification insufficiently described and documented. There were concerns that the methods may have resulted in surprisingly large changes in part as a function of the precise sampling used (some further details relating to this concern are given below).

The original images at 40X magnification have been replaced with images at 100X, and the method of quantification has been clarified:

“Neurolucida (MBF Bioscience, St. Albans, VT) with an Olympus BX61 light microscope was used to trace dendritic morphologies. Using a 40X lens (Olympus Plan N, NA=0.65), dendritic arbors of layer 4 neurons were traced, followed by sholl analysis in Neuroexplorer (MBF Bioscience) to identify the region 75 to 100 μm from soma of basolateral dendrite, which were then traced using a 100X lens (Olympus Plan N Oil, NA=1.25). Dendritic protrusions with lengths 50% greater than the dendritic shaft diameter were identified as spines, and counted in Neuroexplorer.”

3) There is a long list of minor edits indicating many details of the presentation that need clarification or correction.

Each of these have been addressed.

Reviewer #3:Figure 1: label for the panel 'D' is missing from figure. In addition, the data presentation in this panel is not intuitive. Does each data point represent a cell (presumably WFA+)? What are the numbers on each axis (off PV and on PV) represent? How does one WFA+ cell appear to be off PV and on PV at the same time (but to different extent)? The message of this panel is 'the decrease in WFA staining following LRx was observed on and off PV+ cell bodies' (Results paragraph one) but this graph fails to show this right away. Furthermore, does LRx preferentially affect WFA intensity in on PV or off PV population? Please also state how many animals were used in this analysis apart from total cell numbers.

We have added label D, and added the following clarification to the Materials and methods:

“PV^+^ somata were identified by size exclusion (80-300 μm^[2]^) and fluorescence intensity (auto threshold + 30). Region of interest (256 pixels) was defined by a line scan (single pixel height) through the center of a PV+ soma (x, y)=(a, b), WFA intensity was determined for PV^+^ and PV^-^ regions. Control versus LRx (6 subjects, 240 cells versus LRx (4 subjects, 249 cells)*p<0.005, Student’s T-test.”

Figure 3, panel C: it is unclear how the quantification was carried out. In the sample micrograph, two arrows (blue and red) are present but they are not explained in the figure legend. Do they represent 'on PV' and 'off PV' populations of VGluT2 respectively?

Yes and this has been clarified in the legend.

What is the criteria for classifying 'on PV' or 'off PV' VGlut2 puncta?

We have changed the nomenclature to PV^+^ and PV^-^ to denote co-localization with PV immunoreactivity

The use of same colour scheme (red and blue) for Con and LRx groups in quantification below also becomes confusing.

We have revised the colour scheme of the figure.

Also, does LRx increase co-localization between MMP biomarker and VGluT2 in both on PV/off PV populations? The figure legend suggests a 'parallel increase' (Figure 3 legend) but the quantification graphs otherwise show a significant difference in 'off PV' only. The n numbers for this panel are questionable – does the data come from '12, 8 cells' or '6, 7 slices for Con and LRx'?

We have included statistics that demonstrate a significant decrease in both PV^+^ and PV^-^ puncta; N is now expressed as the number of subjects.

Figure 6, panel A: why does MMP-9 KO mice non-deprived eye have very high VEP amplitude (>100uV) at base level (before MD) even in Con (no DE) condition? Given Kelly et al., 2015 show that MMP-9 KO mice have normal ocular dominance levels similar to that of WT mice, this would suggest that MMP-9 KO mice have even higher contralateral response (perhaps >200uV). This contradicts Figure 5—figure supplement 2 which shows much lower VEP responses from both contralateral and ipsilateral eyes of MMP-9 KO mice. In addition, Kelly et al. showed that MMP-9 KO mice showed an OD shift after 7d MD (increase in non-deprived eye response), which does not match with the data presented here. Please explain these discrepancies.

In agreement with Kelly et al., 2015, we observe normal contralateral eye preference for VEPs, although the absolute amplitude of both the contralateral and ipsilateral eye VEP are increased, (Figure 5—figure supplement 2) consistent with increased excitability as we report here and previously (Murase et al., 2016). Importantly, in the experiments depicted in Figure 6 and Figure 7, we occluded the input to the ipsilateral, non-dominant eye, which induces a strengthening of the contralateral, dominant input from the initial value of 125 microvolts.

Figure 6, panel B: Sample images of Golgi stained basolateral dendrites are very blurry and it is hard to appreciate how individual spines can be identified for quantification. Better images with clear labeling of identified spines would be necessary.

The original images at 40X magnification have been replaced with images at 100X, and the method of quantification has been clarified:

“Neurolucida (MBF Bioscience, St. Albans, VT) with an Olympus BX61 light microscope was used to trace dendritic morphologies. Using a 40X lens (Olympus Plan N, NA=0.65), dendritic arbors of layer 4 neurons were traced, followed by sholl analysis in Neuroexplorer (MBF Bioscience) to identify the region 75 to 100 μm from soma of basolateral dendrite, which were then traced using a 100X lens (Olympus Plan N Oil, NA=1.25). Dendritic protrusions with lengths 50% greater than the dendritic shaft diameter were identified as spines, and counted in Neuroexplorer.”

[Editors' note: further revisions were requested prior to acceptance, as described below.]

Please address the remaining points. This should not need to be sent back to reviewers. The manuscript may be considered "accepted pending minor revisions."Reviewer #3:Hyaluronidase non-specificity is the major limitation of this study. But since it can rescue MMP-9 KO phenotype, some common substrates or pathways are suggested to be targeted by these two enzymes. The authors should focus their discussion on potentially shared downstream targets.

We have added the following information regarding the targets of hyaluronidase and MMP-9 to the Discussion:

“In addition, we show that the increase in perisynaptic MMP activity induced by salient experience is associated with a specific class of synapses (thalamic inputs to cortex), and a decrease in aggrecan, a known MMP-9 substrate (Mercuri et al., 2000). Hyaluronidase treatment of cultured neurons also reduces immunoreactivity for aggrecan (Giamanco et al., 2010), suggesting the two enzymes may share common substrates. However degradation of hyaluronic acid, the primary target of hyaluronidase, would also result in reduced aggrecan clustering. Importantly, targets are both lyases are found in throughout diffuse and condensed ECM (Verslegers, Lemmens, Van Hove and Moons, 2013).”